# Triggers of defensive medical behaviours: a cross-sectional study among physicians in the Netherlands

Erik Renkema,[1] Kees Ahaus,[1,2] Manda Broekhuis,[1] Maria Tims[3]

[1]Department of Operations, Faculty of Economics and Business, University of Groningen, Groningen, The Netherlands
[2]Department of Health Services Management & Organisation, Erasmus School of Health Policy & Management, Erasmus University Rotterdam, Rotterdam, The Netherlands
[3]Department of Management and Organisation, School of Business and Economics, Amsterdam Business Research Institute, VU University Amsterdam, Amsterdam, The Netherlands

**Correspondence to**
Dr Erik Renkema;
e.h.renkema@rug.nl

## ABSTRACT

**Objectives** This study investigated whether the attitudes of physicians towards justified and unjustified litigation, and their perception of patient pressure in demanding care, influence their use of defensive medical behaviours.

**Design** Cross-sectional survey using exploratory factor analysis was conducted to determine litigation attitude and perceived patient pressure factors. Regression analyses were used to regress these factors on to the ordering of extra tests or procedures (defensive assurance behaviour) or the avoidance of high-risk patients or procedures (defensive avoidance behaviour).

**Setting** Data were collected from eight Dutch hospitals.

**Participants** Respondents were 160 physicians and 54 residents (response rate 25%) of the hospital departments of (1) anaesthesiology, (2) colon, stomach and liver diseases, (3) gynaecology, (4) internal medicine, (5) neurology and (6) surgery.

**Primary outcome measures** Respondents' application of defensive assurance and avoidance behaviours.

**Results** 'Disapproval of justified litigation' and 'Concerns about unjustified litigation' were positively related to both assurance ($\beta$=0.21, p<0.01, and $\beta$=0.28, p<0.001, respectively) and avoidance ($\beta$=0.16, p<0.05, and $\beta$=0.18, p<0.05, respectively) behaviours. 'Self-blame for justified litigation' was not significantly related to both defensive behaviours. Perceived patient pressures to refer ($\beta$=0.18, p<0.05) and to prescribe medicine ($\beta$=0.23, p<0.01) had direct positive relationships with assurance behaviour, whereas perceived patient pressure to prescribe medicine was also positively related to avoidance behaviour ($\beta$=0.14, p<0.05). No difference was found between physicians and residents in their defensive medical behaviour.

**Conclusions** Physicians adopted more defensive medical behaviours if they had stronger thoughts and emotions towards (un)justified litigation. Further, physicians should be aware that perceived patient pressure for care can lead to them adopting defensive behaviours that negatively affects the quality and safety of patient care.

## INTRODUCTION

Defensive medicine is a widespread approach among physicians that negatively affects healthcare costs[1] and the quality and safety of patient care.[2 3] It is defined as the ordering of extra tests or procedures (assurance behaviour), or the avoidance of

### Strengths and limitations of this study

► In this study a broad spectrum of thoughts and emotions regarding litigation was measured, and a distinction was made between justified and unjustified types of litigation.

► This is the first study in which specific forms of patient pressure have been related to physicians' and residents' defensive medical behaviour.

► A drawback of the study is its reliance on self-reported perceptions and behaviour.

high-risk patients or procedures (avoidance behaviour), primarily to reduce the risk of being held liable for malpractice.[4 5] Studies show that between 60% and 95% of physicians adopt defensive medical behaviour.[4 6–11] The inappropriate ordering of invasive procedures or medicines costs society unnecessary expense[4] and exposes patients to unnecessary health risks[11], while the avoidance of high-risk patients or procedures reduces patients' access to healthcare.[11] Traditionally, the adoption of defensive medical behaviour is explained by concerns about malpractice litigation.[8 9 11–15] However, there is still considerable uncertainty as to how litigation concerns affect defensive medical behaviour. For example, physicians vary in their thoughts and emotions towards litigation. Some regard litigation as a form of justice or as contributing to the quality of care, while others have a more negative attitude that is associated with emotions such as fear, stress, and anger, and a low self-confidence.[16] Despite these differences in how physicians think and feel about litigation and what they consider as justified and unjustified, the extent to which attitude differences play a role in defensive medical behaviour is unknown. Furthermore, physicians' attitudes may differ for litigation which they view as 'justified', for example when they made a mistake or were negligent, and litigation which they view as 'unjustified', for example following an adverse event where

they do not feel they acted incorrectly. A large proportion of malpractice litigation cases can be seen as 'unjustified' since they are dismissed in court[17] with physicians' behaviour not being regarded as negligent.[18] The first objective of this study is to identify whether different attitudes towards justified and unjustified litigation lead to differences in defensive medical behaviour.

This study focuses on thoughts and emotions of self-blame, concerns and disapproval which are for practical reasons labelled as 'negative', although in combination with litigation might reflect attitude aspects that align with the overall aims of the medical malpractice liability system.

Similarly, the aspect of patient pressure, in combination with litigation attitude, has not been addressed in defensive medicine research. Patients have become increasingly demanding in their requests for care, and may even pressurise physicians to provide specific forms of care.[19–21] Perceived pressure from patients is associated with higher referral rates,[22 23] and an increased likelihood of examining patients and over-prescribing drugs.[9 21 24–27] What remains unknown is whether perceived patient pressure strengthens the relationship between litigation attitude and defensive medicine. Consequently, the second objective of this study is to investigate whether perceived patient pressure drives physicians and residents to apply assurance or avoidance behaviours. Besides attending physicians (ie, a person qualified to practice medicine), resident physicians (ie, a medical graduate engaged in specialised practice under supervision in a hospital) were included in the study. Previous research[9 10 28] has found that defensive medicine is also prevalent among residents. Furthermore, some studies have reported that physicians alter their teaching behaviour[29] to encourage their residents to practice defensively[30] in response to the fear of malpractice liability. As such, it seems important to investigate whether litigation attitude and patient pressure play a role in the defensive medical behaviours of residents. Based on this discussion, the following hypotheses have been formulated:

Hypothesis 1: A negative attitude towards justified litigation is positively related to (a) assurance and (b) avoidance behaviours by physicians and residents.

Hypothesis 2: A negative attitude towards unjustified litigation is unrelated to (a) assurance and (b) avoidance behaviours by physicians and residents.

Hypothesis 3: Perceived patient pressure for care strengthens the positive relationship between a negative attitude towards justified litigation and the (a) assurance and (b) avoidance behaviours by physicians and residents.

## METHOD
### Study design and sample
Between September 2014 and March 2015 a cross-sectional survey was conducted to investigate the relation between physicians' and residents' 'negative' attitudes towards justified and unjustified litigation, their perception of patient pressure in demanding care and their use of defensive medical behaviours.

Data were collected from physicians and residents working at eight Dutch hospitals. The hospitals are a mix of general and academic hospitals, geographically spread across the Netherlands. Prior to the data collection, the questionnaire was reviewed with one physician, and a pilot test carried out involving 32 physicians (data not included in the final analysis). The boards of directors, the medical staff or patient safety departments were asked for permission to distribute the questionnaire within their hospital in the departments of (1) anaesthesiology, (2) colon, stomach and liver diseases, (3) gynaecology, (4) internal medicine, (5) neurology and (6) surgery. These specialisms were selected as they are relatively vulnerable to patients' complaints and litigation. After approval from the head of the medical staff of the relevant department(s), an email was sent to the physicians and residents containing a short explanation of the research and a hyperlink to the electronic survey. In the email, and in the questionnaire, it was explicitly stated that all the information provided would be used anonymously. Participation was voluntary. Given that neither patients nor patients' data were involved in the study, ethical approval for the study was not required under the Dutch Law on Medical Research Involving Human Subjects.

### Patient and public involvement
Patients and public were not involved in the development, design, recruitment and randomisation of this study.

### Measures
*Negative attitudes towards justified and unjustified litigation* were measured with 22 statements derived from existing research that probed thoughts and emotions regarding the potential individual consequences of justified and unjustified litigation.[16 31–33] Participants were asked about their emotions (eg, guilt, worry, stress) and thoughts (eg, improve the quality of care, a form of justice (reversed items)) about potential consequences of accusations. Participants were asked to respond to each statement using a 5-point Likert scale ranging from 1 (strongly disagree) to 5 (strongly agree). See online supplementary appendix 1 for the full survey instrument.

*Perceived patient pressure* was measured with 12 statements derived from earlier studies on perceived patient pressure[21 25–27] that were adapted to match the practices of physicians. These statements were divided into three groups of four items, addressing perceived patient pressure to examine, to refer and to prescribe medicine. Respondents rated how frequently they experienced patient pressure with options ranging from 1 (never) to 5 (always).

*Defensive medicine* was measured with seven statements derived from studies on defensive medical behaviour.[10 11] Participants were asked for assurance behaviours, such as prescribing unnecessary medication, ordering tests that are not clinically indicated, carrying out unnecessary procedures and making unnecessary referrals. They were

also asked for avoidance behaviours such as refusing to apply high-risk procedures. One previously used avoidance behaviour statement (ie, refusing to treat high-risk patients) was split into two separate statements (ie, refusing to treat patients with complex medical problems, and refusing to treat patients from whom one expects an accusation) to add clarity. Response options again ranged from 1 (never) to 5 (always).

## Data analysis strategy

Prior to testing the hypotheses, a factor analysis was conducted using SPSS Statistics V.22 (IBM Corporation) with principal axis factoring[34] to examine the psychometric properties of the measures. The criterion for retaining factors was an eigenvalue[35] >1. Items were considered to contribute sufficiently to a factor when their loading[36] was >0.36. Consequently, items that loaded <0.36, or cross-loaded >0.36, were removed during the analysis along with two-item factors since factors containing less than three items are usually weak.[37] To ease interpretation of the factor structure, we applied oblique rotation using direct oblimin and a delta of zero.[38] Cronbach's α values were calculated for the identified factors as well as the mean scores and SD. Listwise deletion was applied in all analyses.

First, separate exploratory factor analyses were conducted on subsets of the data (ie, litigation attitude, perceived patient pressure and defensive medical behaviour) to determine the factor structure that best fitted each subset of items. Based on the separate factor analyses, 14 items were removed from the original data set due to their low factor loadings, high cross-loadings or for being part of a two-item factor. Then the overall factor structure was checked through an exploratory factor analysis of the remaining 27 items (0.785; $\chi^2(351)=2255.868$, p<0.001). One further item was removed during this analysis due to a low factor loading and two more items that made up a two-item factor. Table 1 shows the seven-factor solution that emerged and the corresponding reliability estimates.

In more detail, three factors relate to a physician's or resident's attitude towards litigation: self-blame for justified litigation (three items); disapproval of justified litigation (four items) and concerns about unjustified litigation (three items). Two factors emerged related to perceived patient pressure: perceived patient pressure to refer (three items); and perceived pressure to prescribe medicine (four items).

Defensive medical behaviour emerged as two distinct factors: assurance behaviour (four items); and avoidance behaviour (three items). The correlation between these two behaviours was low (r=0.30; p<0.01).

Each quantitative variable was summarised as a mean value and SD. Next, the correlations between the control variables (gender, age, specialism, workplace, employment form, litigation experience and formal measures after litigation) and the dependent variables (assurance and avoidance behaviours) were examined. This analysis showed that age (>60 years) and specialism

(anaesthesiology) correlated significantly with avoidance behaviour and these were therefore included in the model as control variables. A dummy variable was included for 'grade' in the model to measure potential differences in defensive behaviour between physicians and residents. Following this, the hypotheses were tested with two multiple regression analyses in which litigation attitude and perceived patient pressure were regressed on to assurance and avoidance behaviours. Finally, the interaction between litigation attitude and perceived patient pressure was investigated using a SPSS macro designed by Hayes.[39]

## RESULTS

A total of 238 responses were received. Due to missing data of 24 participants, the study sample consisted of 214 physicians and residents (response rate 25%), which is in line with other studies regarding safety attitudes and medical incidents[40 41] and can be regarded as acceptable given the professionals approached.[42] The characteristics of the respondents are provided in table 2. As shown in the second column, the sample is a good reflection of the target population.

Fourteen respondents were excluded from some of the analyses because they felt unable to answer some of the questions regarding defensive medicine and perceived patient pressure. This was because they regarded prescribing medicine as not a usual procedure within their specialism.

The means (SD) of all variables and the correlations between the variables are displayed in table 3. The analysis revealed that 177 of 200 respondents (89%) claimed to, at least sometimes, apply one or more forms of assurance behaviour, and 90 of 214 respondents (42%) some form of avoidance behaviour.

Means for the perceived patient pressure factors were 1.86 (0.45) and 1.58 (0.48) respectively, and the individual items showed that 191 of 214 respondents (89%) felt, at least sometimes, pressured by their patients to refer, and 144 of 200 respondents (72%) to prescribe medicine. Perceived pressure to examine did not emerge as a distinct factor.

While most factors had an acceptable Cronbach's α (>0.70), the factors related to disapproval of justified litigation and to avoidance behaviour had reliability coefficients slightly below this threshold. The descriptive statistics show no signs of multicollinearity in the factor solution and support the assumption that all factors, except for self-blame for justified litigation, are significantly related to either assurance or avoidance behaviours. Factors identified in the exploratory factor analysis were entered into a multiple linear regression model and evaluated as possible predictors of both assurance and avoidance defensive medicine behaviour.

## Hypothesis testing

The regression analysis (tables 4 and 5) shows that the regression models that included litigation attitude and

**Table 1** Exploratory factor analysis (n=200), Cronbach's α values and factor loadings

| Item wording | Loadings | | | | | | |
|---|---|---|---|---|---|---|---|
| | **SBJL** | **DISJL** | **COUJL** | **PPREF** | **PPMED** | **ASB** | **AVB** |
| α | **0.726** | **0.663** | **0.860** | **0.772** | **0.884** | **0.803** | **0.650** |
| **SBJL** | | | | | | | |
| The potential consequences of justified accusations evoke a feeling of guilt | **0.903** | −0.038 | 0.015 | 0.023 | −0.018 | 0.036 | 0.021 |
| The potential consequences of justified accusations evoke a feeling of shame | **0.760** | −0.046 | −0.072 | 0.029 | −0.004 | 0.035 | 0.096 |
| I regard the potential consequences of justified accusations as a personal attack | **0.372** | 0.111 | −0.057 | −0.136 | −0.018 | −0.092 | −0.195 |
| **DISJL** | | | | | | | |
| I regard the potential consequences of a justified accusation as a form of justice (reversed) | −0.098 | **0.643** | −0.016 | −0.126 | 0.018 | 0.094 | 0.015 |
| The potential consequences of justified accusations are justified (reversed) | −0.064 | **0.612** | 0.018 | 0.024 | −0.138 | −0.053 | 0.019 |
| The potential consequences of justified accusations are a way to improve the quality of care (reversed) | 0.054 | **0.561** | 0.098 | 0.089 | 0.131 | 0.158 | −0.080 |
| The potential consequences of justified accusations make me feel angry | 0.111 | **0.485** | −0.117 | −0.045 | 0.055 | −0.039 | −0.005 |
| **COUJL** | | | | | | | |
| The potential consequences of unjustified accusations evoke in me the fear of being accused | −0.022 | −0.028 | **−0.855** | 0.043 | −0.023 | 0.057 | 0.070 |
| The potential consequences of unjustified accusations worry me | 0.028 | −0.040 | **−0.778** | 0.020 | −0.019 | 0.048 | −0.106 |
| The potential consequences of unjustified accusations make me feel stressed | 0.122 | 0.090 | **−0.745** | −0.070 | 0.072 | −0.009 | −0.007 |
| **PPREF** | | | | | | | |
| When a patient directly or indirectly criticises previous treatments, I feel pressured to refer the patient even if it is clinically not strictly necessary | −0.018 | 0.029 | −0.037 | **−0.761** | 0.024 | 0.036 | −0.044 |
| When a patient quotes decisions of other physicians to stress the need for intervention, I feel pressured to refer the patient even if it is clinically not strictly necessary | 0.025 | −0.064 | −0.009 | **−0.743** | −0.102 | −0.026 | −0.055 |
| When a patient requests further examination or treatment, I feel pressured to refer the patient even if it is clinically not strictly necessary | 0.070 | 0.126 | 0.046 | **−0.546** | −0.175 | 0.069 | 0.131 |
| **PPMED** | | | | | | | |
| When a patient describes their symptoms in extensive emotional words, I feel pressured to prescribe medicine even if it is clinically not strictly necessary | 0.022 | 0.004 | −0.008 | 0.055 | **−0.800** | −0.035 | −0.089 |
| When a patient quotes decisions of other physicians to stress the need for intervention, I feel pressured to prescribe medicine even if it is clinically not strictly necessary | −0.020 | 0.002 | −0.023 | −0.083 | **−0.789** | 0.087 | 0.044 |
| When a patient requests further examination or treatment, I feel pressured to prescribe medicine even if it is clinically not strictly necessary | −0.013 | 0.009 | −0.014 | −0.079 | **−0.785** | 0.027 | 0.068 |
| When a patient directly or indirectly criticises previous treatments, I feel pressured to prescribe medicine even if it is clinically not strictly necessary | 0.040 | −0.036 | 0.056 | −0.054 | **−0.764** | 0.092 | −0.060 |
| **ASB** | | | | | | | |
| How often do you prescribe unnecessary medication to prevent a potential accusation? | 0.039 | 0.037 | 0.098 | 0.108 | −0.095 | **0.768** | 0.028 |
| How often do you order tests that are not clinically indicated in order to prevent a potential accusation? | −0.025 | −0.007 | −0.157 | −0.088 | −0.021 | **0.638** | −0.058 |

**Table 1** Continued

| Item wording | Loadings | | | | | | |
|---|---|---|---|---|---|---|---|
| | SBJL | DISJL | COUJL | PPREF | PPMED | ASB | AVB |
| α | **0.726** | **0.663** | **0.860** | **0.772** | **0.884** | **0.803** | **0.650** |
| How often do you carry out procedures that are probably unnecessary in order to prevent a potential accusation? | 0.017 | 0.024 | −0.126 | 0.008 | −0.063 | **0.626** | −0.104 |
| How often do you make unnecessary referrals to physicians with other specialisms to prevent a potential accusation? | −0.016 | 0.079 | −0.081 | −0.241 | 0.028 | **0.595** | −0.037 |
| AVB | | | | | | | |
| How often do you refuse to treat patients with complex medical problems in order to avoid a potential accusation in the event of a complication | −0.041 | 0.108 | −0.093 | 0.190 | −0.126 | −0.074 | **−0.659** |
| How often do you refuse to treat patients from whom you expect an accusation in order to prevent a potential accusation in the event of a complication | 0.019 | −0.068 | 0.025 | −0.060 | 0.081 | 0.112 | **−0.604** |
| How often do you avoid high-risk procedures in order to prevent a potential accusation in the event of a complication | −0.012 | 0.013 | −0.002 | −0.092 | −0.040 | 0.043 | **−0.592** |
| Eigenvalues | **1.516** | **2.235** | **1.098** | **1.250** | **3.124** | **5.295** | **1.771** |
| % of variation explained (cumulative) | **6.315** | **15.628** | **20.205** | **25.412** | **38.428** | **60.489** | **67.870** |

Bold values indicates items that loaded >0.36.
ASB, assurance behaviour; AVB, avoidance behaviour; COUJL, concerns about unjustified litigation; DISJL, disapproval of justified litigation; PPMED, perceived patient pressure to prescribe medicine; PPREF, perceived patient pressure to refer; SBJL, self-blame for justified litigation.

perceived patient pressures (model 5) were significant both for assurance behaviour ($F(13,186)=5.817$, p<0.001) and for avoidance behaviour ($F(13,186)=3.540$, p<0.001).

In line with hypothesis 1a, 'disapproval of justified litigation' was positively related to assurance behaviour (β=0.21, p<0.01). However, no significant relationship was found between 'self-blame for justified litigation' and assurance behaviour (β=−0.06, p=0.42). Turning to hypothesis 1b, 'disapproval of justified litigation' (β=0.16, p<0.05) was positively related to avoidance behaviour, providing support for hypothesis 1b, whereas 'self-blame for justified litigation' was not (β=0.07, p<0.35). The negative attitudes towards justified litigation explained 9% of the variance in assurance behaviour and 6% of the variance in avoidance behaviour.

In hypothesis 2, it was suggested that a negative attitude towards unjustified litigation would be unrelated to (a) assurance and (b) avoidance behaviours. However, results showed that concerns about unjustified litigation were positively related to both assurance (β=0.28, p<0.001) and avoidance behaviours (β=0.18, p<0.05), leading us to reject hypothesis 2 (a and b). In addition, a negative attitude towards unjustified litigation explained 8% of variance in assurance behaviour and 2% of the variance in avoidance behaviour.

Finally, the interaction between perceived patient pressure and negative litigation attitudes was tested in the prediction of assurance (hypothesis 3a) and avoidance (hypothesis 3b) behaviours. None of the interaction terms was significant. However, interestingly, perceived patient pressures to refer (β=0.18, p<0.05) and to prescribe medicine (β=0.23, p<0.01) had direct positive relationships with assurance behaviour, explaining an additional 11% of the variance in assurance behaviour. In terms of avoidance behaviour, only perceived patient pressure to prescribe medicine was positively related to avoidance behaviour (β=0.14, p<0.05).

## DISCUSSION AND CONCLUSIONS

The goal of this study was to investigate physicians' and residents' attitudes towards justified and unjustified litigation in relation to defensive medical behaviour, and whether perceived patient pressures strengthen these relationships. Despite the low mean scores for assurance and avoidance behaviours, the vast majority of the respondents in this study did admit to engaging in assurance and avoidance behaviours (89% and 42%, respectively). These findings of higher prevalence of assurance behaviour are in line with previous studies.[4 6–11]

Those physicians and residents who disapproved of justified litigation were more likely to employ assurance and avoidance behaviours. A possible explanation for their disapproval is the criminalisation of errors, through which mistakes can be judged as negligence, which might lead to litigation.[43] No relationship was found between self-blame for justified litigation and physicians' defensive medical behaviour. This was unexpected since previous research has found that the perception of personal failure following a complaint causes shame that induces physicians to adopt defensive medicine.[44] However, this study reveals that concerns about unjustified litigation

**Table 2** Characteristics of the survey participants (n=214) and target population[57]

| Variables | No. of respondents (%) | Target population (%) |
|---|---|---|
| Gender | | |
| Male | 121 (56.5) | 5882 (55.0) |
| Female | 93 (43.5) | 4804 (45.0) |
| Age (years) | | |
| <30 | 22 (10.3) | |
| 30–40 | 68 (31.8) | |
| 41–50 | 64 (29.9) | |
| 51–60 | 50 (23.4) | |
| >60 | 10 (4.7) | |
| Specialism | | |
| Anaesthesiology | 55 (25.7) | 2307 (21.6) |
| Colon, stomach and liver | 19 (8.9) | 734 (6.9) |
| Gynaecology | 29 (13.6) | 1436 (13.4) |
| Internal medicine | 43 (20.1) | 3148 (29.5) |
| Neurology | 20 (9.3) | 1295 (12.1) |
| Surgery | 34 (15.9) | 1766 (16.5) |
| Other | 14 (6.5) | |
| Grade | | |
| Physician | 160 (74.8) | 7936 (74.3) |
| Resident | 54 (25.2) | 2750 (25.7) |
| Workplace | | |
| General hospital | 132 (61.7) | |
| Academic hospital | 78 (36.4) | |
| Other workplace | 4 (1.9) | |
| Employment form | | |
| Employed | 119 (55.6) | |
| Self-employed | 95 (44.4) | |
| Litigation experience | | |
| Yes | 34 (15.9) | |
| No | 180 (84.1) | |
| Formal measures after litigation (of n=34 with litigation experience) | | |
| Yes | 4 (11.8) | |
| No | 30 (88.2) | |

**Table 3** Means, SD and correlations among the study variables

| Variable | M (SD) | 1 | 2 | 3 | 4 | 5 | 6 | 7 | 8 | 9 |
|---|---|---|---|---|---|---|---|---|---|---|
| 1. Age (1=over 60 years; 0=60 years or younger) | 0.05 (0.21) | | | | | | | | | |
| 2. Anaesthesiologist (1=Anaest.; 0=Other spec.) | 0.26 (0.44) | 0.072 | | | | | | | | |
| 3. Physician (1=Physician; 0=Resident) | 0.75 (0.44) | 0.129 | 0.071 | | | | | | | |
| 4. SBJL | 3.31 (0.76) | −0.148* | −0.064 | −0.030 | | | | | | |
| 5. DISJL | 2.51 (0.59) | −0.011 | 0.068 | −0.041 | 0.092 | | | | | |
| 6. COUJL | 3.98 (0.77) | 0.025 | 0.140* | 0.017 | 0.439** | 0.139* | | | | |
| 7. PPREF | 1.86 (0.45) | 0.051 | −0.177** | 0.006 | 0.223** | 0.154* | 0.215** | | | |
| 8. PPMED | 1.58 (0.48) | 0.016 | −0.060 | 0.073 | 0.040 | 0.041 | 0.038 | 0.412** | | |
| 9. ASB | 1.76 (0.57) | 0.072 | 0.041 | 0.057 | 0.134 | 0.286** | 0.332** | 0.350** | 0.315** | |
| 10. AVB | 1.26 (0.38) | 0.236** | 0.225** | 0.039 | 0.104 | 0.219** | 0.281** | 0.111 | 0.131 | 0.297** |

The means are the averages of participants' responses on the underlying items (range: 1–5).

*Significant at the 0.05 level (two-tailed); **significant at the 0.01 level (two-tailed).

ASB, assurance behaviour; AVB, avoidance behaviour; COUJL, concerns about unjustified litigation; DISJL, disapproval of justified litigation; PPMED, perceived patient pressure to prescribe medicine; PPREF, perceived patient pressure to refer; SBJL, self-blame for justified litigation.

seem to trigger avoidance and particularly assurance behaviours. It may be that physicians and residents associate unjustified litigation with stressful long-lasting litigation procedures[45] and negative media attention,[46] even if they are eventually cleared of any blame. This suggests that physicians and residents strongly fear becoming the 'second victim'[15 47 48] of an adverse event through unjustified litigation. This finding might also suggest that they fear being held accountable for what in their eyes is an

**Table 4** Results of regression analysis for assurance behaviour

| Predictor | β (95% CI) | | | | | |
|---|---|---|---|---|---|---|
| | Model 1 (R²=0.01) | Model 2 (R²=0.01) | Model 3** (R²=0.10) | Model 4*** (R²=0.18) | Model 5*** (R²=0.28) | Model 6*** (R²=0.29) |
| Age>60 (vs. age<60) | 0.07 (−0.07 to 0.21) | 0.06 (−0.08 to 0.21) | 0.09 (−0.05 to 0.23) | 0.06 (−0.07 to 0.20) | 0.05 (−0.08 to 0.17) | 0.04 (−0.09 to 0.17) |
| Anaesthesiology (vs. other specialisms) | 0.04 (−0.11 to 0.18) | 0.03 (−0.12 to 0.18) | 0.02 (−0.12 to 0.16) | −0.03 (−0.17 to 0.11) | 0.01 (−0.12 to 0.14) | 0.01 (−0.13 to 0.14) |
| Grade | | | | | | |
| Physician (vs. resident) | | 0.05 (−0.10 to 0.19) | 0.04 (−0.10 to 0.17) | 0.03 (−0.10 to 0.16) | 0.02 (−0.11 to 0.14) | 0.02 (−0.11 to 0.15) |
| Attitude towards justified litigation | | | | | | |
| SBJL | | | 0.12 (−0.02 to 0.25) | −0.03 (−0.17 to 0.12) | −0.06 (−0.20 to 0.08) | −0.06 (−0.21 to 0.09) |
| DISJL | | | 0.27 (.14 to 0.40)*** | 0.25 (0.12 to 0.37)*** | 0.21 (0.09 to 0.33)** | 0.21 (0.08 to 0.33)** |
| Attitude towards unjustified litigation | | | | | | |
| COUJL | | | | 0.31 (0.17 to 0.46)*** | 0.28 (0.14 to 0.42)*** | 0.28 (0.13 to 0.43)*** |
| Perceived patient pressure | | | | | | |
| PPREF | | | | | 0.18 (0.03 to 0.33)* | 0.19 (0.04 to 0.34)* |
| PPMED | | | | | 0.23 (0.09 to 0.36)** | 0.22 (0.08 to 0.36)** |
| Interactions | | | | | | |
| SBJL × PPREF | | | | | | 0.02 (−0.14 to 0.18) |
| SBJL × PPMED | | | | | | 0.03 (−0.14 to 0.19) |
| DISJL × PPREF | | | | | | 0.00 (−0.14 to 0.15) |
| DISJL × PPMED | | | | | | −0.03 (−0.19 to 0.13) |
| COUJL × PPREF | | | | | | −0.03 (−0.19 to 0.13) |
| COUJL × PPMED | | | | | | 0.01 (−0.17 to 0.16) |

*P<0.05; **P<0.01; ***P<0.001.
COUJL, concerns about unjustified litigation; DISJL, disapproval of justified litigation; PPMED, perceived patient pressure to prescribe medicine; PPREF, perceived patient pressure to refer; SBJL, self-blame for justified litigation.

**Table 5** Results of regression analysis for *avoidance* behaviour

| Predictor | β (95% CI) | | | | | |
|---|---|---|---|---|---|---|
| | Model 1*** (R²=0.13) | Model 2*** (R²=0.13) | Model 3*** (R²=0.19) | Model 4*** (R²=0.21) | Model 5*** (R²=0.23) | Model 6*** (R²=0.24) |
| Age>60 (vs. age<60) | 0.26 (0.12 to 0.39)*** | 0.25 (0.12 to 0.39)*** | 0.28 (0.15 to 0.41)*** | 0.27 (0.13 to 0.40)*** | 0.27 (0.14 to 0.40)*** | 0.26 (0.13 to 0.40)*** |
| Anaesthesiology (vs. other specialisms) | 0.24 (0.10 to 0.37)** | 0.24 (0.10 to 0.37)** | 0.23 (0.10 to .34)** | 0.20 (0.07 to .34)** | 0.21 (0.07 to 0.34)** | 0.20 (0.06 to 0.34)** |
| Grade | | | | | | |
| Physician (vs. resident) | | 0.01 (−0.12 to 0.14) | 0.00 (−0.13 to 0.13) | 0.00 (−0.13 to 0.13) | −0.01 (−0.14 to 0.11) | 0.00 (−0.14 to 0.12) |
| Attitude towards justified litigation | | | | | | |
| SBJL | | | 0.15 (0.02 to 0.28)* | 0.07 (−0.08 to 0.21) | 0.07 (−0.08 to 0.21) | 0.07 (−0.08 to 0.22) |
| DISJL | | | 0.17 (0.04 to 0.30)** | 0.16 (0.03 to 0.28)*** | 0.16 (0.03 to 0.28)* | 0.15 (0.02 to 0.28)* |
| Attitude towards unjustified litigation | | | | | | |
| COUJL | | | | 0.18 (0.04 to 0.33)*** | 0.18 (0.04 to 0.33)* | 0.17 (0.01 to 0.32)* |
| Perceived patient pressure | | | | | | |
| PPREF | | | | | −0.04 (−0.19 to 0.11) | −0.04 (−0.19 to 0.12) |
| PPMED | | | | | 0.14 (0.00 to 0.28)* | 0.12 (−0.03 to 0.26) |
| Interactions | | | | | | |
| SBJL × PPREF | | | | | | −0.04 (−0.20 to 0.13) |
| SBJL × PPMED | | | | | | 0.13 (−0.04 to 0.30) |
| DISJL × PPREF | | | | | | −0.02 (−0.17 to 0.14) |
| DISJL × PPMED | | | | | | −0.03 (−0.19 to 0.14) |
| COUJL × PPREF | | | | | | 0.01 (−0.15 to 0.17) |
| COUJL × PPMED | | | | | | −0.01 (−0.18 to 0.16) |

*P<0.05; **P<0.01; ***P<0.001.
COUJL, concerns about unjustified litigation; DISJL, disapproval of justified litigation; PPMED, perceived patient pressure to prescribe medicine; PPREF, perceived patient pressure to refer; SBJL, self-blame for justified litigation.

unjustified litigation case, and this induces them to adopt defensive medical behaviours. If this is indeed the case, then a valuable step would be to find a way to prevent physicians and residents having to go through unjustified litigation proceedings. This could perhaps be achieved by improving the filtering out of unjustified cases at an early stage to prevent them ending up in court.

Unlike in earlier studies,[10 11 28] this study did not reveal significant statistical differences between physicians and residents in either their negative attitudes towards litigation or their defensive medical behaviour. This finding might be explained by a recent study[30] that reported that medical residents frequently encounter defensive medicine practices and are being taught to take malpractice liability into consideration when making clinical decisions. The findings regarding disapproval of justified litigation and concerns about unjustified litigation indicate which thoughts and emotions may be being passed on to residents during their practical training. As such, finding ways to prevent these thoughts and emotions being instilled during academic medical training is key to reducing defensive medical behaviour.

It was further expected that perceived patient pressure would be a catalyst in the relationship between litigation attitude and defensive medical behaviour. However, as no interaction effects were found, this catalytic role was not clearly evident. A possible reason for this is that both the means and the SD of perceived patient pressures were low, making it harder to detect a significant interaction.[49] Nevertheless, direct relationships between perceived patient pressure and both assurance and avoidance behaviours were identified. That is, physicians and residents who more often perceive pressure to refer or to prescribe medicine more often demonstrate assurance behaviours than their colleagues who perceive these pressures less frequently. This finding may indicate that physicians and residents 'give in' to the demands of their patients in order to avoid conflicts, even if this goes against their professional judgement. Using assurance behaviours may be a way to prevent a good relationship with the patient going sour. Indeed, the preservation of a good doctor–patient relationship has been suggested elsewhere as a potential explanation for defensive medical behaviour.[9 50]

Although perceived pressure to prescribe medicine was related to avoidance behaviour, it did not add significantly to the explained variance. This indicates that physicians and residents rarely respond to perceived patient pressures by refusing to treat patients or avoiding certain procedures, maybe because they do not see this as an appropriate response. The relationship that was found between perceived patient pressures and defensive medical behaviour suggests that residents might be 'trained' to recognise patient pressures and accordingly act defensively. This would imply that it is important to provide training in the patient safety curriculum on communication skills to deal with perceived patient pressures and to not let these influence one's clinical decision-making. Physicians and residents should be made aware that their perceptions of patient pressures can lead them to adopt assurance and avoidance behaviours.

For respondents older than 60 years, there was a significant relationship with avoidance behaviour. An explanation for this finding may be found in the selection optimisation theory.[51 52] This theory states that people, when they are older, prefer to undertake less complicated and risky tasks than younger people. Our findings are in line with a study by Marin et al,[53] showing more defensive practices among older doctors, though some scholars found that younger physicians show more defensive medicine behaviour.[10] Therefore more research is needed to conclude on the relationship between age and defensive medicine behaviour.

Anaesthesiologists were more inclined to refuse high-risk patients and procedures than the other specialists we included in our study. With its long-standing and strong focus on reducing patient harm, anaesthesiology is seen as the specialism that provides the standard for patient safety.[54] Therefore, an explanation for our finding might be that the strong focus on reducing patient harm has translated into a greater emphasis on risk-averse behaviour by anesthesiologists than by those in other specialisms. Future research is needed to see whether this relationship is confirmed elsewhere.

A strength of this study is that a broad spectrum of thoughts and emotions regarding litigation was measured, and a distinction was made between justified and unjustified types of litigation. Another strength of this study is that it is the first in which specific forms of patient pressure have been related to defensive medical behaviour by physicians and residents, showing that perceptions of patient pressure are crucial in determining the adoption of defensive medical behaviours. A third strength is that we were able to collect the responses of 200 physicians and residents on the delicate topic of defensive behaviours, and that this sample turned out to be an adequate representation of the targeted population. Nevertheless, this study has several limitations. First, our data were self-reported perceptions and behaviours. Although there are no obvious concerns with the data in that variables are distinguishable and correlations between them are not excessive, other data sources could be used to verify the outcomes of this study. The second limitation is the study's cross-sectional design, which makes it impossible to determine the causality of relationships. Nevertheless, the literature[55 56] suggests that attitude is an important driver of behaviour and supports the direction of the relationships that were hypothesised between the theoretically linked variables. Future researchers are encouraged to use longitudinal studies to further investigate the relationships we found. Third, the research was executed in a context in which there is a relatively modest litigation climate. Given that behaviour is context-specific, it is reasonable to expect the relationships found in this

study for defensive medical behaviour to be stronger in more litigious climates.

Another promising area for future research is care settings outside of hospitals. As general practitioners often refer patients to hospitals for further investigation, primary care might for example be an interesting setting to investigate the relationships we found.

To summarise, this study demonstrates that physicians' and residents' 'disapproval of justified litigation' and their 'concerns about unjustified litigation' stimulate both assurance and avoidance forms of defensive medical behaviour, whereas 'self-blame for justified litigation' did not. Further, perceptions of patient pressure to refer and to prescribe medicine stimulate physicians and residents to adopt both avoidance and assurance behaviours that can negatively affect the quality and safety of care. To prevent physicians implicitly instilling defensive medical behaviour in residents, medical curricula should specifically address litigation attitudes and how to deal with perceived patient pressures.

**Acknowledgements** The authors would like to thank all physicians and residents who participated in their study.

**Contributors** ER designed the study, collected, analysed and interpreted the data, and drafted the manuscript. KA co-designed the study, made significant contributions to the data acquisition and the interpretation of the data, and revised the manuscript. MB co-designed the study, made significant contributions to the analysis and interpretation of the data, and revised the manuscript. MT co-designed the study, contributed to the analysis and interpretation of the data, and revised the manuscript.

**Funding** The authors have not declared a specific grant for this research from any funding agency in the public, commercial or not-for-profit sectors.

**Competing interests** None declared.

**Patient consent for publication** Not required.

**Provenance and peer review** Not commissioned; externally peer reviewed.

**Data sharing statement** No additional unpublished data from the study are available.

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
