## [Reviewer comments · BMJ Open]

ARTICLE DETAILS

TITLE (PROVISIONAL)	Triggers of defensive medical behaviors. A cross-sectional study among physicians in the Netherlands.
AUTHORS	Renkema, Erik; Ahaus, Kees; Broekhuis, Manda; Tims, Maria

VERSION 1 – REVIEW

REVIEWER	Carmela Rinaldi University of Eastern Piedmont, Via Solaroli 17, Novara, Italy And A.O.U. Maggiore della Carità, Novara, Italy
REVIEW RETURNED	13-Jul-2018

GENERAL COMMENTS	The paper addresses a relevant topic. Investigated whether the attitudes of physicians and residents towards justified and unjustified litigation, and their perception of patient pressure in demanding care, influence their use of defensive medical behaviors. Some revisions are required: 1. For general content: in a scientific article, one should not use the first singular or plural person (I or We). E.g. use: a cross-sectional study was conducted ...Not we measured...2. Page 4, Introduction, lines 15-27: I suggest adding more recent references to studies carried out in Europe. E.g.: PMID 28534429 - PMID 273735793. Page 4, Introduction, lines 49-50: how can we define a litigation justified and unjustified? Please define a litigation justified and unjustified.4. Page 5, Introduction, lines 3- 33 I suggest adding more recent references5. Page 6, Method: better to write "Study design and sample"6. Page 6, Method: more information on the type of study is needed7. Page 6, Method: did the participants sign an informed consent?8. Page 6, method: better to specify for points inclusion criteria and exclusion criteria9. Page 6, from lines 39 down, these and table 1 are Results not method10. Page 7, description table 1, better if: Table1. Characteristics of the survey participants (N 214) Variables N (%) Still in the table 1: Formal measures after litigation (N 34)11. Page 8, Measures, lines 24-45: it would be useful to show, as attachment, the two collection tools (12 statements and 7 statements).
---

	12. Page 8, Data Analysis Strategy: specify better analysis type. did you conduct analysis with logistic regression model first univariate and then multivariate? 13. Results: the contents are good, but confusing. They should respond to points for research / hypothesis questions. 14. Page 9: Results. You write “We first conducted separate exploratory factor analyses on subsets of the data (i.e. litigation attitude, perceived patient pressure and defensive medical behavior) to determine the factor structure that best fitted each subset of items. Based on the separate factor analyses, we removed 14 items from the original dataset due to their low factor loadings, high crossloadings, or for being part of a two-item factor. We then checked the overall factor structure through an exploratory factor analysis of the remaining 27 items (0.785; $\chi^2(351) = 2255.868, p < .001$). One further item was removed during this analysis due to a low factor loading and two more items that made up a two-item factor. Table 2 shows the seven-factor solution that emerged and the corresponding reliability estimates.”. All very specific and well written. However, this should be explained in "Methods"
--	--

REVIEWER	Ciallella Costantino Legal Medicine - SISMLA Department Sapienza University of Rome
REVIEW RETURNED	26-Jul-2018

GENERAL COMMENTS	Review: Why is defensive behavior so deeply rooted in medicine? A cross – sectional study among physicians and residents in the Netherlands Authors collected data from eight Dutch Hospitals studying how attitudes towards justified/unjustified litigation and perceived pressures from patients can influence medical choices about treatments and exams prescription. This is a new approach to the defensive medicine problem: paper can be accepted. The study is well and clearly designed and its subject – defensive medicine - is an important and actual topic to investigate, mainly for the negative influence it has on healthcare costs and safety of patient care. The approach is new because the Authors also considered physicians’ perceptions about litigations and patients’ behavioral pressures and correlated them to the defensive medical behaviors adopted. About the survey: response rate was 25% which is statistically significant; Authors used existing researches and studies to measure “attitudes towards justified and unjustified litigation”, “perceived patient pressure” and “defensive medicine”, making personal changes. It would be appropriate to spend a few words to explain the statements from the earlier study. About the results: at first, the Authors conducted separate analyses on subsets of data, removing items with low factor loadings. Final items were seven, collected in Table 2. The Authors showed Descriptive Statistics, Results of regression analysis for
--

	assurance behavior and for avoidance behavior in tables 3, 4 and 5. It's a good work, no corrective measures are needed. To conclude the article can be published.
--	--

VERSION 1 – AUTHOR RESPONSE

Response to comments

Manuscript title: *Why is defensive behavior so deeply rooted in medicine? A cross-sectional study among physicians and residents in the Netherlands*

We would like to thank the editor and the reviewers for their encouraging reviews and for the careful reading of our manuscript and detailed suggestions for improvements.

Point-by-point response to comments of reviewer 1 in the decision letter

No	Comments and answers to comments
General remark	The paper addresses a relevant topic. Investigated whether the attitudes of physicians and residents towards justified and unjustified litigation, and their perception of patient pressure in demanding care, influence their use of defensive medical behaviors.
Response	Thank you.
C1	For general content: in a scientific article, one should not use the first singular or plural person (I or We). E.g. use: a cross-sectional study was conducted ...Not we measured...
Response	We removed the terms 'I', 'We' and 'Our' and replaced them by a general formulation of the sentences.
C2	Page 4, Introduction, lines 15-27: I suggest adding more recent references to studies carried out in Europe. E.g.: PMID 28534429 – PMID 27373579
Response	The following more recent references were added to this section and to the discussion section: Asher et al., 2013 Panella et al., 2016; Panella et al., 2017; Schiess et al., 2018. We adjusted the percentage of physicians that adopt defensive medical behavior, based on what was reported in Panella et al. (2017) to between 60% and 95%.
C3	Page 4, Introduction, lines 49-50: how can we define a litigation justified and unjustified? Please define a litigation justified and unjustified.
Response	Whether litigation is justified or unjustified is dependent on how physicians perceive it. A sentence on page 4 was added to emphasize this. The following text, provided on page 4, provides examples of how litigation might be viewed as justified and unjustified: "Furthermore, physicians' attitudes may differ for litigation which they view as 'justified', e.g. when they made a mistake or were negligent, and litigation which they view as 'unjustified', e.g. following an adverse event where they do not feel they acted incorrectly."
C4	Page 5, Introduction, lines 3- 33 I suggest adding more recent references
Response	The following more recent references were added to this section: Lewis and Tully, 2011; Fenton et al., 2015; Ringberg et al, 2014; Fletcher-Lartey et al., 2016.

C5	Page 6, Method: better to write "Study design and sample"
Response	We replaced the subtitle by "Study design and sample".
C6	Page 6, Method: more information on the type of study is needed
Response	We added the following sentence at the start of the method section: "Between September 2014 and March 2015 a cross-sectional survey was conducted to investigate the relation between physicians' and residents' attitudes towards justified and unjustified litigation, their perception of patient pressure in demanding care, and their use of defensive medical behaviors."
C7	Page 6, Method: did the participants sign an informed consent?
Response	Participants did not sign an informed consent. However, in the communication towards the hospitals as well as the individuals that were approached to participate in the study, it was stated that the information they provided would be used anonymously. They were also informed about the study's purpose. Furthermore participation was voluntary.
C8	Page 6, method: better to specify for points inclusion criteria and exclusion criteria.
Response	We added the following sentence on the inclusion of the selected specialisms in the study on page 6: "These specialisms were selected as they are relatively vulnerable to patients' complaints and litigation." This was based on published data of the Dutch Disciplinary Committee regarding the numbers of physicians, that were involved in litigation, by specialism.
C9	Page 6, from lines 39 down, these and table 1 are Results not method.
Response	These lines and table 1 have been transferred from the Method to the Results section (pages 12-14 of the revised manuscript) .
C10	Page 7, description table 1, better if: Table1. Characteristics of the survey participants (N 214) Variables N (%) Still in the table 1: Formal measures after litigation (N 34)
Response	The suggested changes were incorporated to make table 1 more clear .
C11	Page 8, Measures, lines 24-45: it would be useful to show, as attachment, the two collection tools (12 statements and 7 statements).
Response	All statements are now included as an attachment: appendix 1 'Full survey instrument'.
C12	Page 8, Data Analysis Strategy: specify better analysis type. did you conduct analysis with logistic regression model first univariate and then multivariate?
Response	The following sentence was added (on page 16 of the revised manuscript) to specify better the type of analysis used: " Factors identified in the exploratory factor analysis were entered into a multiple linear regression model and evaluated as possible predictors of both assurance and avoidance defensive medicine behavior."
C13	Results: the contents are good, but confusing. They should respond to points for research / hypothesis questions.
Response	As suggested by the reviewer in comment 14, we replaced the description of the exploratory factor analysis including the accompanying table from the Results to the Method section (pages 8-12 of the revised manuscript).
C14	Page 9: Results. You write "We first conducted separate exploratory factor analyses on subsets of the data (i.e. litigation attitude, perceived patient pressure and defensive medical behavior) to determine the factor structure that best fitted each subset of items. Based on the separate factor analyses, we removed 14 items from the original dataset due to their low factor loadings, high cross loadings, or for being part of a two-item factor. We then checked the overall factor structure through an exploratory factor analysis of the

	remaining 27 items (0.785; $\chi^2(351) = 2255.868, p < .001$). One further item was removed during this analysis due to a low factor loading and two more items that made up a two-item factor. Table 2 shows the seven-factor solution that emerged and the corresponding reliability estimates.”. All very specific and well written. However, this should be explained in "Methods"
Response	We replaced this text, including the table, from the Results to the Method section (pages 8-12 of the revised manuscript).

Point-by-point response to comments of reviewer 2 in the decision letter

No	Comments and answers to comments
General remark	The paper is interesting, well organized and with clear and useful results. This is a new approach to the defensive medicine problem: paper can be accepted.
Response	Thank you.
C1	It would be appropriate to spend a few words to explain the statements from the earlier studies.
Response	We have added to the Methods section a few sentences to explain the statements from earlier studies.

VERSION 2 – REVIEW

REVIEWER	Carmela Rinaldi University of Eastern Piedmont, Via Solaroli 17, Novara, Italy
REVIEW RETURNED	26-Oct-2018

GENERAL COMMENTS	The authors rewrote adequately the paper by adhering to the numerous revisions I asked. However, for greater accuracy, I ask for a second expert opinion on the statistical analysis shown in Tables 1, 3, 4 and 5. After this review I would say that the paper can be published.
---

REVIEWER	Andrew Hinde University of Southampton, United Kingdom
REVIEW RETURNED	20-Dec-2018

GENERAL COMMENTS	I was not one of the reviewers of the previous version of this paper, but I have been asked to look at the revised version. I have read the comments of the reviewers of the previous version and the authors' response. Here are my comments on this version. p. 2, ll.13-14. It might be helpful to readers to have the brief definitions of 'assurance' and 'avoidance' behaviours in the abstract as well as in the text. You could replace 'defensive assurance and avoidance behaviours' by 'the ordering of extra tests or procedures (defensive assurance behaviour) or the
--

	avoidance of high-risk patients or procedures (defensive avoidance behaviour)'. p. 5, l. 11. Explain the difference between 'attending physicians' and 'resident physicians'. p. 5, l. 20. What proportion of physicians have a positive attitude towards unjustified litigation? Perhaps it is the strength of the negative attitude that is being measured, rather than whether or not the attitude is negative? p. 6, ll. 1-3. I would test hypothesis 3 by stratifying on the basis of perceived patient pressure for care and examining the relationship between attitudes towards justified litigation and assurance and avoidance behaviours within each stratum. You could divide your sample into two roughly equal strata to achieve this. This is a simpler test than using interaction effects in a regression model. Did you consider doing this? You could easily do it. p. 12, l. 22. You received 238 responses not '238 surveys'! p. 12, l. 23 - p. 13, l. 2. The response rate of 25% is quite low (even though it may be in line with other similar surveys). You do have some basic demographic information, presented in Table 2, p.13, which you could use to check whether your respondents are representative of all physicians and residents in the target population. I should like to see some analysis of this type. It need only involve adding another column to Table 2 giving the percentages in the target population for those variables for which you have this information. Table 3, p. 15. Could you say what the statistics are in the main body of the table? 'Descriptive statistics' is too vague a designation. In the note you say what the means are, but you give no details about the numbers in the columns headed 'Var1' to 'Var9'. I guess they are correlation coefficients of some kind. But you have a mixture of categorical and continuous variables, including three associations (between Var1, Var2 and Var3) which are based on a 2 x 2 contingency table. You can compute correlation coefficients in such cases, but chi-squared or similar tests are more usually applied. Table 5, p. 18. I thought that the positive associations between age and anaesthesiology and avoidance behaviour were worth a comment in the text. I find the age effect plausible. Doctors who are close to retirement are likely to be disinclined to take on complex or high-risk cases. For them, life is too short for that. p. 21, ll. 14-16. An additional point is that you have only around 200 cases. 
--	---

REVIEWER	Harm Van Marwijk Brighton and Sussex Medical School, United Kingdom
REVIEW RETURNED	03-Jan-2019

GENERAL COMMENTS	This in itself interesting study idea investigated whether the attitudes of Dutch physicians and residents towards justified and unjustified litigation and their perception of patient pressure in demanding care influenced their use of defensive medical behaviours. I like their courage to take up this delicate area of
--

	research. As this is a highly sensitive area, it is, however, essential to get the tone right or this type of study could be more harmful than helpful to the profession (or the authors should not try to publish in a medical journal). The fact is that legal matters influence medicine, but that might be to some extent inevitable. Risk tolerance is not a simple matter and is determined, in the UK, for instance, by people having open access to A&Es. Working in the UK as a former Dutch and now UK GP, I am afraid that their sample of physicians may not be the best to research this question. The authors cannot help this, obviously, but it might limit the external validity of their findings to place outside Holland. My indemnity premium in the UK is about ten times as high as in the Netherlands. Defensive medicine is probably happening much more in countries (such as the UK) with a more 'legalised' culture in medicine, and with less public trust in the profession (and scandal-based newspapers) than in the Netherlands. See the effects of the Shipman enquiry. No similar doctor was convicted in the Netherlands, see the recent Tromp case (in which the ruling was reversed). A second important consideration is that in countries with a two-tiered medical system such as the Netherlands and the UK, the highest levels of medical uncertainty or risk-taking are inherently found outside hospitals. The referral system means that higher levels of medical certainty (which is where are the expensive tools are) are expressly expected to be sought in hospitals, mainly to rule out physical diseases. They are inherently defensive. So, why not include primary care in the sampling of doctors? A third consideration in this sensitive area is that there seems to have been little involvement of actual doctors in the design of this project. Methodologically, this means a more hermeneutic approach to the plan. The first sentence 'defensive medicine is a widespread approach' will put the hairs up on the back of most doctors, I am afraid. The title, 'Why is defensive behavior so deeply rooted in medicine? A cross-sectional study among physicians and residents in the Netherlands.' suggests that the authors will give us an answer but that is not forthcoming. I suggest they go for a non-medical journal with this message. It seems to me to be better to study more what defensive medicine actually is, first, and perhaps ask doctors more about what that means. I suggest reading Danczak A et al. Managing uncertainty... A final consideration is that there does not seem to be a validated assessment tool. In short, I would suggest the authors rethink their project, make it a lot less ambitious and seek collaboration with doctors/universities from a country with a higher level of defensive medicine than Holland.
--	--

VERSION 2 – AUTHOR RESPONSE

Response to comments

Manuscript title: *Why is defensive behavior so deeply rooted in medicine? A cross-sectional study among physicians and residents in the Netherlands*

We would like to thank the editor and the reviewers for their encouraging reviews and for the careful reading of our manuscript and detailed suggestions for improvements.

Point-by-point response to comments of reviewer 1 in the decision letter

No	Comments and answers to comments
C1	The authors rewrote adequately the paper by adhering to the numerous revisions I asked. However, for greater accuracy, I ask for a second expert opinion on the statistical analysis shown in Tables 1, 3, 4 and 5. After this review I would say that the paper can be published.
Response	Thank you. Reviewer 3 has given the second expert opinion on the statistical analysis. A point-by-point response to the comments of reviewer 3 can be found below.

Reviewer 2

Response	Reviewer 2 already accepted the previous version of our manuscript. Thank you.
----------	--

Point-by-point response to comments of reviewer 3 in the decision letter

No	Comments and answers to comments
General remark	I was not one of the reviewers of the previous version of this paper, but I have been asked to look at the revised version. I have read the comments of the reviewers of the previous version and the authors' response. Here are my comments on this version.
Response	We would like to thank this reviewer for reviewing our manuscript, the previous version and the comments of the reviewers on the previous version.
C1	p. 2, ll.13-14. It might be helpful to readers to have the brief definitions of 'assurance' and 'avoidance' behaviours in the abstract as well as in the text. You could replace 'defensive assurance and avoidance behaviours' by 'the ordering of extra tests or procedures (defensive assurance behaviour) or the avoidance of high-risk patients or procedures (defensive avoidance behaviour)'.
Response	We have replaced the text and added the brief definition of 'assurance' and 'avoidance' behaviors to the abstract as well.
C2	p. 5, l. 11. Explain the difference between 'attending physicians' and 'resident physicians'.
Response	A 'resident physician' is a medical graduate under supervision, whereas an 'attending physician' is graduated and qualified to practice medicine. We have added the definitions of both physicians in the text.
C3	p. 5, l. 20. What proportion of physicians have a positive attitude towards unjustified litigation? Perhaps it is the strength of the negative attitude that is being measured, rather than whether or not the attitude is negative?
Response	We indeed measure the strength of the negative attitude towards justified and unjustified litigation, and not whether an attitude is positive or negative. We have better indicated this in the abstract and text. (Also see table 1 for the identified variables).
C4	p. 6, ll. 1-3. I would test hypothesis 3 by stratifying on the basis of perceived patient pressure for care and examining the relationship between attitudes towards justified litigation and assurance and avoidance behaviours within each stratum. You could divide your sample into two roughly equal strata to achieve this. This is a simpler test than using interaction effects in a regression model. Did you consider doing this? You could easily do it.
Response	We agree that stratification is simpler than moderation. However, moderation analysis has several important features that make it more valuable to use. Most relevant, with a moderation analysis, we get a parameter estimate and p-value of the difference, allowing

	a statistical test of the difference between high and low experiences of patient pressure. Stratification may also unnecessarily attenuate multicollinearity, especially given that our sampling design was not based on stratification of perceived patient pressure.
C5	p. 12, l. 22. You received 238 responses not '238 surveys'!
Response	Thank you. We have corrected 'surveys' by 'responses' in the text.
C6	p. 12, l. 23 - p. 13, l. 2. The response rate of 25% is quite low (even though it may be in line with other similar surveys). You do have some basic demographic information, presented in Table 2, p.13, which you could use to check whether your respondents are representative of all physicians and residents in the target population. I should like to see some analysis of this type. It need only involve adding another column to Table 2 giving the percentages in the target population for those variables for which you have this information.
Response	We have included a second row in the table providing demographics for all physicians and residents that, in our opinion, show that our respondents are representative for the target population.
C7	Table 3, p. 15. Could you say what the statistics are in the main body of the table? 'Descriptive statistics' is too vague a designation. In the note you say what the means are, but you give no details about the numbers in the columns headed 'Var1' to 'Var9'. I guess they are correlation coefficients of some kind. But you have a mixture of categorical and continuous variables, including three associations (between Var1, Var2 and Var3) which are based on a 2 x 2 contingency table. You can compute correlation coefficients in such cases, but chi-squared or similar tests are more usually applied.
Response	We changed the header of the table (to better explain its content) into: "Means, standard deviations and correlations among the study variables". The statistics in the table indeed represent correlation coefficients. We agree with the reviewer that correlations are not the most informative statistics when working with categorical variables. However, it is still informative to see whether the categorical variables correlate with the continuous study variables. It is therefore quite common to present these statistics in a correlation table, as well given that the correlation table functions as a descriptive analysis and does not represent the substantive test of the study hypotheses.
C8	Table 5, p. 18. I thought that the positive associations between age and anaesthesiology and avoidance behaviour were worth a comment in the text. I find the age effect plausible. Doctors who are close to retirement are likely to be disinclined to take on complex or high-risk cases. For them, life is too short for that.
Response	We agree with the reviewer that the findings concerning age and anesthesiology and avoidance behavior are worth to be commented. Therefore we added on page 22 several comments to the discussion about the positive association between age and anesthesiology and avoidance behavior.
C9	p. 21, ll. 14-16. An additional point is that you have only around 200 cases.
Response	We think that 200 cases is a substantial amount considering the sensitivity of the topic of defensive behavior in the field of medicine. We added the following sentence on page 23 to state this: "A third strength is that we were able to collect the responses of 200 physicians and residents on the delicate topic of defensive behaviors, and that this sample turned out to be an adequate representation of the targeted population."

Point-by-point response to comments of reviewer 4 in the decision letter

No	Comments and answers to comments
General remark	This in itself interesting study idea investigated whether the attitudes of Dutch physicians and residents towards justified and unjustified litigation and their perception of patient pressure in demanding care influenced their use of defensive medical behaviours. I like

	their courage to take up this delicate area of research. As this is a highly sensitive area, it is, however, essential to get the tone right or this type of study could be more harmful than helpful to the profession (or the authors should not try to publish in a medical journal). The fact is that legal matters influence medicine, but that might be to some extent inevitable. Risk tolerance is not a simple matter and is determined, in the UK, for instance, by people having open access to A&Es.
Response	Thank you for addressing the relevance of this research topic. We agree that this is a delicate topic. In our paper we refer to previous research which has indeed shown the effect of legal matters on medicine. It is not our intention to be harmful to the profession but instead to better understand under which condition defensive medicine occurs. For example, the finding that perceived patient pressure and disapproval of justified litigation are associated with defensive behaviors, indicate ways in which physicians can be better supported by policy or training. We carefully worded our findings and the discussion of our findings.
C1	Working in the UK as a former Dutch and now UK GP, I am afraid that their sample of physicians may not be the best to research this question. The authors cannot help this, obviously, but it might limit the external validity of their findings to place outside Holland. My indemnity premium in the UK is about ten times as high as in the Netherlands. Defensive medicine is probably happening much more in countries (such as the UK) with a more 'legalised' culture in medicine, and with less public trust in the profession (and scandal-based newspapers) than in the Netherlands. See the effects of the Shipman enquiry. No similar doctor was convicted in the Netherlands, see the recent Tromp case (in which the ruling was reversed).
Response	We agree with the reviewer that the litigation context differs per country. Nevertheless we show in this research that even in a country with a modest litigation climate like Holland, the percentage of physicians reporting to apply defensive behavior is comparable to other countries. Furthermore, we show that it is related to litigation attitude and perceived patient pressure. With regards to the generalization of our findings we have mentioned a limitation on page 23: "Third, the research was executed in a context in which there is a relatively modest litigation climate. Given that behavior is context-specific, it is reasonable to expect the relationships found in this study for defensive medical behavior to be stronger in more litigious climates."
C2	A second important consideration is that in countries with a two-tiered medical system such as the Netherlands and the UK, the highest levels of medical uncertainty or risk-taking are inherently found outside hospitals. The referral system means that higher levels of medical certainty (which is where the expensive tools are) are expressly expected to be sought in hospitals, mainly to rule out physical diseases. They are inherently defensive. So, why not include primary care in the sampling of doctors?
Response	We agree with the author that it would be highly interesting to research defensive behaviors within primary care as well, however the purpose of the current study was to investigate the relationships towards defensive behaviors of physicians within hospitals. We have added a future suggestion to investigate the relationships we found in primary care settings on page 23: "Another promising area for future research are care settings outside of hospitals. As general practitioners often refer patients to hospitals for further investigation, the primary care might for example be an interesting setting to investigate the relationships we found."
C3	A third consideration in this sensitive area is that there seems to have been little involvement of actual doctors in the design of this project. Methodologically, this means a more hermeneutic approach to the plan. The first sentence 'defensive medicine is a widespread approach' will put the hairs up on the back of most doctors, I am afraid. The title, 'Why is defensive behavior so deeply rooted in medicine? A cross-sectional study among physicians and residents in the Netherlands.' suggests that the authors will give

	us an answer but that is not forthcoming. I suggest they go for a non-medical journal with this message. It seems to me to be better to study more what defensive medicine actually is, first, and perhaps ask doctors more about what that means. I suggest reading Danczak A et al. Managing uncertainty...
Response	We agree with the reviewer that the title of the manuscript does not provide the best description of its content. Therefore we changed the title to: "Triggers of defensive medical behaviors. A cross-sectional study among physicians in the Netherlands." We would like to point out that we can draw on a lot of experience with physicians in this study. The study in the manuscript is part of a PhD research project of the first author. Prior to this study many in-depth interviews have been conducted with physicians about their thoughts and emotions regarding litigation and the disclosure of incidents. With the knowledge and experience gathered from these interviews we carefully developed the survey for the current study. The questions were developed by the authors of which two of them have more than twenty years of research experience within the medical sector. The questionnaire that was set up for this study was reviewed by one physician and a pilot test was carried out involving 32 physicians. The fact that defensive medicine is a widespread approach has been shown in many studies that have been published in medical journals like JAMA (e.g. Studdert et al., 2005), Social Science & Medicine (e.g. He, 2014) or Archives of Internal Medicine (e.g. Bishop et al., 2010), which we refer to in our manuscript. In this respect we think it is appropriate and valuable when our manuscript is published in a medical journal as it contributes to the explanation of defensive behaviors within medicine.
C4	A final consideration is that there does not seem to be a validated assessment tool.
Response	Defensive medicine has been studied substantially and we have made use of the questions similar to the studies of Studdert et al. (2005) and Ortashi et al. (2013). For litigation attitude and perceived patient pressure we had to develop our own assessment tools, as to our knowledge no assessment tools are available thus far. We conducted exploratory factor analyses that revealed relevant, valid, and reliable factors which we used in our regression analyses.
C5	In short, I would suggest the authors rethink their project, make it a lot less ambitious and seek collaboration with doctors/universities from a country with a higher level of defensive medicine than Holland.
Response	As mentioned in our answer to the first comment of the reviewer, we do think that Holland has been an interesting context to research the triggers of defensive medical behaviors. Furthermore, we want to stress that we have sought collaboration with doctors as we have explained in our answer to comment 3 of the reviewer.

VERSION 3 – REVIEW

REVIEWER	Andrew Hinde University of Southampton, United Kingdom
REVIEW RETURNED	05-Mar-2019
GENERAL COMMENTS	Thank you for your comprehensive and persuasive response to my report on the previous version. I think this version of the paper should be published. It is an interesting and important contribution to the factors associated with the practice of defensive medicine and should stimulate further research into the subject.

	The extra column in Table 2 clearly demonstrates that your sample, though modest in size for the reasons you give, is suitably representative of the population from which it is drawn. This makes your analysis more convincing.
--	---

REVIEWER	Harm Van Marwijk Brighton and Sussex Medical School, United Kingdom
-----------------	--

REVIEW RETURNED	18-Mar-2019
-------------

GENERAL COMMENTS	Nice rebuttal
---------------